# *Thymus zygis,* Valuable Antimicrobial (In Vitro and In Situ) and Antibiofilm Agent with Potential Antiproliferative Effects

**DOI:** 10.3390/plants12233920

**Published:** 2023-11-21

**Authors:** Milena D. Vukić, Natália Čmiková, Anis Ben Hsouna, Rania Ben Saad, Stefania Garzoli, Marianna Schwarzová, Nenad L. Vuković, Ana D. Obradović, Miloš M. Matić, Bożena Waszkiewicz-Robak, Maciej Ireneusz Kluz, Miroslava Kačániová

**Affiliations:** 1Department of Chemistry, Faculty of Science, University of Kragujevac, 34000 Kragujevac, Serbia; milena.vukic@pmf.kg.ac.rs (M.D.V.);; 2Institute of Horticulture, Faculty of Horticulture and Landscape Engineering, Slovak University of Agriculture, Tr. A. Hlinku 2, 94976 Nitra, Slovakiamarianna.schwarzova@gmail.com (M.S.); 3Laboratory of Biotechnology and Plant Improvement, Centre of Biotechnology of Sfax, B.P “1177”, Sfax 3018, Tunisiaraniabensaad@gmail.com (R.B.S.); 4Department of Chemistry and Technologies of Drug, Sapienza University, P. le Aldo Moro, 5, 00185 Rome, Italy; stefania.garzoli@uniroma1.it; 5Department of Biology and Ecology, Faculty of Science, University of Kragujevac, 34000 Kragujevac, Serbia; 6School of Medical & Health Sciences, University of Economics and Human Sciences in Warsaw, Okopowa 59, 01 043 Warszawa, Poland; b.waszkiewicz-robak@vizja.pl (B.W.-R.); m.kluz@vizja.pl (M.I.K.)

**Keywords:** *T. zygis*, essential oil, chemistry, antimicrobial activity, antiproliferative effects

## Abstract

With the growing issues of food spoilage, microbial resistance, and high mortality caused by cancer, the aim of this study was to evaluate *T. zygis* essential oil (TZEO) as a potential solution for these challenges. Here, we first performed GC/MS analysis which showed that the tested TZEO belongs to the linalool chemotype since the abundance of linalool was found to be 38.0%. Antioxidant activity assays showed the superiority of TZEO in neutralizing the ABTS radical cation compared to the DPPH radical. The TZEO was able to neutralize 50% of ABTS^•+^ at the concentration of 53.03 ± 1.34 μg/mL. Antimicrobial assessment performed by employing disc diffusion and minimal inhibitory concentration assays revealed TZEO as a potent antimicrobial agent with the highest inhibition activity towards tested gram-negative strains. The most sensitive on the treatment with TZEO was *Enterobacter aerogenes* showing an MIC 50 value of 0.147 ± 0.006 mg/mL and a MIC 90 value of 0.158 ± 0.024 mg/mL. Additionally, an in situ analysis showed great effects of TZEO in inhibiting gram-negative *E. coli*, *P. putida*, and *E. aerogenes* growing on bananas and cucumbers. Treatment with the TZEO vapor phase in the concentration of 500 μg/mL was able to reduce the growth of these bacteria on the food models to the extent > 90%, except for *E. coli* growth on the cucumber, which was reduced to the extent of 83.87 ± 4.76%. Furthermore, a test on the antibiofilm activity of the tested essential oil revealed its biofilm prevention effects against *Salmonella enterica* which forms biofilms on plastic and stainless-steel surfaces. Performed tests on the TZEO effects towards cell viability showed no effects on the normal MRC-5 cell line. However, the results of MTT assay of TZEO effects on three cancer cell lines (MDA-MB-231, HCT-116, and K562) suggest that TZEO exerted the strongest effects on the inhibition of the viability of MDA-MB-231 cells, especially after long-term treatment in the highest concentration applied with reducing the viability of the cells to 57%. Additionally, results of NBT and Griess assays suggest that TZEO could be a convenient candidate for future testing for developing novel antitumor therapies.

## 1. Introduction

The main issues for the food industry sector are pathogenic bacteria and foodborne spoilage. These concerns largely affect food quality and safety and contribute to global health problems [1]. According to the WHO (World Health Organization), almost 1 in 10 people worldwide become ill from contaminated food, and 420,000 people die each year from the same cause. Moreover, 110 billion US$ is lost each year in productivity and medical expenses caused by hazardous food, mainly in low- and middle-income countries. Another concerning factor is the ability of bacteria to easily transfer from planktonic to sessile mode. This transition allows them to survive and grow longer on different surfaces and strengthens the resistance of the bacteria against standard antibiotic treatment [2,3]. The formation of this organized multicellular assemblage of microorganisms, or biofilm, increases the chance of pathogen contamination of processed food products [3,4,5,6]. In the struggle with the presented issue of preventing food spoilage, a commonly employed technique in industrial processes was the application of synthetic antimicrobial agents and chemical food additives [7]. However, consumer demand for minimally processed and preservative-free products prompted this industry sector to turn to healthier solutions. Considering this, scientists, industrial research, and development sectors worldwide have focused on studying natural herbs and plant-based biopreservatives that can replace synthetic ones [7,8]. Moreover, the use of naturally obtained preservers has numerous advantages. For example, they can be produced at a low cost, they exert no side effects, their production and use are eco-friendly, and they are locally available.

Although the development of novel technologies has made great progress in medicine, cancer still represents one of the most persistent diseases with high rates of mortality [9]. Due to the emergence of chemo-resistant tumor morphologies, the efficacy of traditional chemotherapies has decreased. Considering this, the need for novel treatment combinations with greater cytotoxicity against malignant cells and minimal adverse effects on healthy tissues has increased. In addition, free radicals are one of the key factors involved in tumor progression. However, its overproduction can lead to a negative clinical impact on therapy by causing chemoresistance. Therefore, the discovery of new and effective sources of natural constituents has attracted the attention of the scientific community, despite the recent progress in the synthesis of new anticancer agents [10]. This fact is confirmed by studies revealing that naturally occurring compounds such as terpenoids, phenolics, flavonoids, and alkaloids exert significant antitumor potential [11].

As one of the plant’s highly valued products, essential oils (EOs) represent the diverse mixtures of compounds with proven beneficial biological effects. These compounds are formed by different biogenetic pathways in all plant organs. Mostly, volatile compounds presented in EOs belong to the classes of terpenes, terpenoids, phenolics, phenylpropanoids, etc. [12,13]. In industrial branches, they are primarily employed as flavoring agents in the manufacturing of different beverages or food products, but they have also found application in the cosmetics industry, in agriculture, and in aromatherapy [14,15,16,17,18].

*Thymus zygis*, belonging to the Lamiaceae family and the *Thymus* genus, represents a taxonomically complex group of plants [19,20,21]. Based on the chemometric investigation, this species includes the seven main chemotypes, including thymol, and carvacrol which, according to literature data, are the most common, followed by linalool, α-terpinyl acetate, thymol/p-cymene/γ-terpinene, 1,8-cineole/myrcene/spathulenol, and 1,8-cineole/α-terpineol [19]. Traditionally, this plant has been used as medicine, but has also been used as a spice [20,22]. Recent studies of different chemotypes of *T. zygis* essential oils have demonstrated their beneficial antimicrobial, antibiofilm, anti-inflammatory, antiseptic, antioxidant, and antispasmodic effects [20,23,24,25,26,27,28]. 

Considering the current urgency for new and natural preservatives and therapeutics, as well as the demonstrated pharmacological effects of different species of *T. zygis* essential oil (TZEO) here, its chemical composition, antioxidant activity, antimicrobial effects (in vitro and in situ), antibiofilm activity, toxicity towards human lung normal fibroblast cell line, and antiproliferative effects have been assessed. Chemical composition was assessed using GC/MS analysis, whereas the antioxidant capacity of TZEO was evaluated by employing DPPH and ABTS assays. Initially, the antimicrobial effects were evaluated on the three Gram-positive and Gram-negative bacteria, and four yeast species using the disc diffusion and minimal inhibitory concentration methods. The antimicrobial effects of TZEO vapor phase were examined in situ on banana and cucumber as food models infected with the same microbial and yeast strains used for the preliminary antimicrobial evaluation. Moreover, driven by the issues regarding biofilm formation threatening effects, herein we have examined the influence of TZEO on the biofilm formation of G^−^ *Salmonella enterica*. Biocompatibility testing was performed using the MTT assay while antiproliferative effects of TZEO were evaluated on three cancer cell lines by using MTT, NBT, and Griess assays. The presented protocols were selected with the aim of providing an in-depth understanding of the TZEO as a beneficial and safe agent in food preservation and its potential toxic effect on different cancer cells. 

## 2. Materials and Methods

### 2.1. Essential Oils

The essential oil (EO) tested in this study was purchased from Hanus s.r.o. (Nitra, Slovakia) and was prepared by distillation of *T. zygis* partially dried cloves, originating from Spain. Throughout the analysis, it was stored in the dark at 4 °C.

### 2.2. GC and GC/MS Analysis

The volatile compounds’ composition of *T. zygis* essential oil (TZEO) was determined by using Agilent Technologies (Palo Alto, Santa Clara, CA, USA) 6890 N gas chromatograph. Volatiles were separated on the HP-5MS capillary column (30 m × 0.25 mm × 0.25 µm). The gas chromatograph was interfaced with a quadrupole mass spectrometer 5975 B (Agilent Technologies, Santa Clara, CA, USA). Agilent Technologies gas chromatograph was operated by HP Enhanced ChemStation software (Agilent Technologies, version number of the software: D.03.00.611). An injection volume was adjusted at 1 µL (10% solution of EO in hexane), with helium 5.0 used as a carrier gas with a 1 mL/min flow rate. The temperature of the split/splitless injector, MS source, and MS quadruple was kept at 280 °C, 230 °C, and 150 °C, respectively. The split ratio was 20:1, and the mass scan range was 35–550 amu at 70 eV. The oven temperature was programmed as follows: 60 °C to 260 °C (increasing rate, 3 °C/min), and the total run time was 67 min. The solvent delay time was 3.20 min for oil sample analysis, while in the case of n-alkanes (C7–C35) it was 2.10 min in order to obtain the retention index for n-heptane (found at 2.6 min). 

A comparison of retention indices (RI) of compounds identified in the TZEO sample with retention indices of n-alkanes (C7–C35) series was used for the identification of the volatiles [29,30]. Moreover, the identification of the volatiles was confirmed by comparing their spectral data with the reference spectra reported in the literature and stored in the MS library (Wiley7Nist) that is merged with the HP Enhanced ChemStation software. Semi-quantification of each component was performed on GC-FID (using the same HP-5MS capillary column), considering amounts higher than 0.1%.

### 2.3. Antioxidant Activity

The antioxidant activity of the TZEO was examined by using two common DPPH and ABTS assays. For the analysis, the stock solution of DPPH^•^ (Sigma Aldrich, Schnelldorf, Germany) in the concentration of 0.025 g/L in methanol (Uvasol^®^ for spectroscopy, Merck, Darmstadt, Germany) was diluted to the absorbance of 0.8 at 515 nm [31]. The ABTS^•+^ was generated according to the previously described procedure and diluted to an absorbance value of 0.7 at 744 nm prior to the analysis [32]. The prepared solutions of DPPH radical and ABTS radical cation were added to a 96-well microtiter plate in a volume of 190 μL. Additionally, plates were supplied with TZEO in the volume of 10 μL in the final concentration range from 3 mg/mL to 0.0235 mg/mL (in methanol). The blank solution used was methanol, and as a standard reference the substance Trolox was dissolved in methanol in the final concentration, which was obtained in the wells in the range of 3–0.015 mg/mL. The reaction mixtures were incubated in the dark at room temperature for 30 min with shaking (1000 rpm), after which the absorbance was measured at 515 nm for DPPH assay and at 744 nm for ABTS assay using a microplate reader. All measurements were performed in triplicate. The results are presented as the total radical scavenging capacity for both assays, expressed according to the Trolox (TEAC) and IC_50_ values. The results are presented as mean values ± standard deviation (SD) of three independent measurements.

### 2.4. Antimicrobial Assays

#### 2.4.1. Microorganisms

The following bacteria were employed in our tests to measure the antibacterial potency of the tested EO: Gram-positive (G^+^) *Listeria monocytogenes* CCM 4699, *Micrococcus luteus* CCM 732, *Staphylococcus aureus* CCM 3953, Gram-negative (G^−^) *Enterobacter aerogenes* CCM 2531, *Escherichia coli* CCM 3954, *Pseudomonas putida* CCM 7156, Yeasts *Candida albicans* CCM 8186, *Candida glabrata* CCM 8270, *Candida krusei* CCM 8271, and *Candida tropicalis* CCM 8223. The Czech collection of microorganisms in Brno, Czech Republic, provided all G^+^ and G^−^ bacterial species and yeasts for this study. The biofilm-forming G^−^ *Salmonella enterica* was isolated and sequenced from milk production for the evaluation of antibiofilm activity. Prior to analysis, the bacterial inoculums were grown in Mueller Hinton Broth (MHB, Oxoid, Basingstoke, UK) for 24 h at 37 °C, whereas yeast inoculums were grown in Sabouraud Dextrose Broth (SDB, Oxoid, Basingstoke, UK) for 24 h at 25 °C. On the day of the experiment, the optical density of the bacterial and yeast inoculums employed was fixed at 0.5 of the McFarland standard.

#### 2.4.2. Disc Diffusion Method

The aforementioned microbial strains were used in the disk-diffusion susceptibility test. Prepared bacterial strains in MHB (0.1 mL) were inoculated to Mueller Hinton Agar (MHA; Merck, Darmstadt, Germany), whereas yeast strains in SDB (0.1 mL) were inoculated to Sabouraud Dextrose Agar (SDA; Merck, Darmstadt, Germany). After being soaked with 10 µL of the tested EO, 6 mm diameter blank discs were placed on the agar surface. While yeasts were incubated at 25 °C for 24 h, bacteria were incubated at 37 °C. The 24 h incubation period was followed by a measurement of the inhibitory activity and the results are presented in mm. For G^−^ and G^+^ bacteria, the well-known antibiotics (ATB) Gentamicin and Cefoxitin (30 µg/disc, Oxoid, Basingstoke, UK) were employed as positive controls, and Fluconazole (30 µg/disc, Oxoid, Basingstoke, UK) was utilized as a positive control for yeasts. The experiment was carried out three times.

#### 2.4.3. MIC Assay

Using the previously described method, the minimal inhibitory concentration values (MIC 50 and MIC 90) were calculated [33]. Briefly, 50 μL of microbe inoculum was added to a microtiter plate containing 96-well. Next, EO in different concentrations (10 mg/mL to 0.00488 mg/mL in MHB) was added in the same volume. Wells containing MHB and SDB with EO (at the proper concentration) were prepared for the negative control, while wells containing MHB and SDB with inoculum were prepared for the maximal growth positive control. Following preparation, prepared plates were incubated for 24 h at 37 °C for bacterial strains and for yeast strains at 25 °C throughout the same period of time. Following this, the content in wells was thoroughly mixed. Prepared plates were next incubated for 24 h at 37 °C for bacterial strains, whereas for yeast strains incubation was performed at 25 °C in the same time interval. Finally, absorbance was measured at 570 nm using a spectrophotometer (Glomax, Promega Inc., Madison, WI, USA). The results are presented as MIC 50 values, or the minimal concentration of the *T. zygis* EO (TZEO) able to inhibit 50% of bacterial growth, and as MIC 90 values, or the minimal concentration of the *T. zygis* EO (TZEO) able to inhibit 90% of bacterial growth. The test was repeated three times.

### 2.5. In Situ Analysis on a Food Model

The selected G^+^ and G^−^ bacteria and yeast strains were used for the in situ assessment of the TZEO antimicrobial effects. For this purpose, commercially available food products were used as bacterial and yeast growth substrates, i.e., banana and cucumber. The applied technique has already been published [34]. Briefly stated, banana and cucumber were washed in distilled water, sliced into 0.5 mm pieces, and dried. After that, bacteria inoculum was applied to the prepared substrates on agar in the 60 mm Petri dishes. The tested EO sample was dissolved in ethyl acetate in concentrations of 500, 250, 125, and 62.5 mg/L and applied to the sterile filter paper. Filter sheets treated only with ethyl acetate were used as a control. The petri dish was closed after filter paper containing the treatment or control was placed on the lid and allowed for a minute to allow the residual ethyl acetate to evaporate. The next step was a 7 day incubation at 37 °C using prepared petri dishes. In situ tests were conducted to measure bacterial growth using stereotypical approaches. The National Institutes of Health, Bethesda, Maryland, USA, provided ImageJ software 1.53t to calculate the volume density (vv) of the bacterial colonies. The volume density of the bacterial colonies was estimated as follows: vv (%) = P/p
where P stands for the points of the stereological grid hitting the colonies, whereas p represents the points of the stereological grid falling to the reference space (growth substrate used). 

The effects of the EOs vapor phase were presented as the percentage (%) of bacterial growth inhibition (BGI):BGI = [(C − T)/C] × 100 (1)
where C corresponds to the control group while T signifies the treatment group. Both groups represent bacterial growth expressed as *v*/*v*. Results obtained as the negative values correspond to growth stimulation.

### 2.6. Biofilm Development Assay

The protein degradation process during the formation of biofilm was evaluated using MALDI-TOF MicroFlex (Bruker Daltonics, Billerica, MA, USA). For this experiment, the 50 mL polypropylene tubes were augmented with 20 mL of MHB and 100 μL of biofilm-forming bacterial inoculum of *S. maltophilia*. Microscopic slides of plastic and stainless steel were next placed in the supplemented tubes. The experimental tubes contained TZEO in the final concentration of 0.1%, while the untreated tubes served as a control. Prepared polypropylene tubes were left on a shaker (170 rpm) and incubated at 37 °C for 3, 5, 7, 9, 12, and 14 days. On each day of the experiment, with a sterile cotton swab, formed biofilms were taken from the tested surfaces (plastic and stainless steel) and directly placed on the target plate. Moreover, planktonic cells from the control samples (without EO) were also tested. Firstly, 300 µL of the control sample culture medium containing a bacterial suspension was centrifuged (13,400× *g*) for 1 min. The obtained pellet was rinsed in ultra-pure water (three times) and was centrifuged again under the same conditions. For the testing, planktonic cells (pellet) were reconstituted in ultrapure water and added to the target plate in the volume of 1 μL. Further, the matrix (1 μL of 10 mg/mL of α-cyano-4-hydroxycinnamic acid) was applied to the plate in the places where the samples of resuspended pellet and swabs were added. A dried plate was placed in MALDI-TOF, and spectra were recorded. The protein spectral data were obtained with the mass-to-charge ratio set at 200–2000 in linear positive mode. By using the automatic analysis, 19 standard global spectrums (MSP) were generated. Generated MSPs were used to create dendrograms using Euclidean Distance Formula [35].

### 2.7. Examination of TZEO Antiproliferative and Redox Status Effects in Cancer Cells

#### 2.7.1. Cell Cultivation and Treatment

To carry out this experiment, the human lung normal fibroblast cell line MRC-5, breast cancer cell line MDA-MB-231, colon cancer cell line HCT-116, and chronic myelogenous leukemia cell line K562 were obtained from the American Tissue Culture Collection. Cells were propagated and maintained in DMEM supplemented with 10% FBS and a combination of antibiotics (100 IU/mL penicillin and 100 µg/mL streptomycin). The cells were grown in a 75 cm^2^ culture flask and supplied with 15 mL DMEM at a confluence of 70% to 80%. The cells were seeded in a 96-well microplate (10,000 cells per well) and cultured in a humidified atmosphere with 5% CO_2_ at 37 °C. After 24 h of cell incubation, 100 μL of medium containing various doses of treatment (1 µg/mL to 200 µg/mL) was added to each well of the microplate, and the cells were incubated for 24 h and 72 h, after which the evaluation of cell viability, superoxide anion radical, and nitrites were measured. The essential oil of *T. zygis* was used in experiments, whereas non-treated cells were used as control. All experiments were performed in triplicate.

#### 2.7.2. MTT Assay

The viability of the cells was determined using MTT assay [36]. Briefly, after a period of incubation, prepared solutions of TZEO in a volume of 100 µL were added to each well. Treated and control cells (cultured only in a medium) were again incubated for 24 h and 72 h. Subsequently, the cell viability was determined by adding 20 µL of MTT (5 mg/mL) to each well. At the end of the reaction, the formed crystals were dissolved in 20 µL of DMSO. The formed color was measured on an ELISA reader at a wavelength of 550 nm. The percentage of viable cells was calculated as the ratio between the absorbance at each dose of the treatment and the absorbance of the non-treated control.

#### 2.7.3. Determination of Superoxide Anion Radical (NBT Assay)

The concentration of superoxide anion radical (O_2_^•–^) was determined using a well-known spectrophotometric method [37]. Prepared cells were treated as explained in the previous section and an assay was performed by adding 20 μL of 5 mg/mL NBT to each well, followed by the cell incubation. Incubation was performed for 1 h at 37 °C in 5% CO_2_. Next, formazan was solubilized in 20 μL DMSO. The absorbances were measured on an ELISA reader at 550 nm. The concentrations of O_2_^•–^ were expressed as nanomoles per milliliter (nmol NBT/mL) in 10^5^ cells.

#### 2.7.4. Determination of Nitrites (Griess Assay)

The spectrophotometric determination of nitrites (NO_2_^–^) as an indicator of the nitric oxide (NO) level was performed by using the Griess method [38]. Equal volumes of 0.1% N-1-napthylethylenediamine dihydrochloride and 1% *w*/*v* sulfanilamide solution in 5% phosphoric acid were mixed to form the Griess reagent immediately before application to the plate. During the 10 min of incubation (at room temperature, protected from the light sources), a purple color developed. After incubation, absorbances were measured on an ELISA reader at 550 nm and the nitrite concentration was expressed in μmol NO_2_^–^/mL in 10^5^ cells.

## 3. Results and Discussion

### 3.1. GC and GC/MS Analysis of TZEO Volatile Composition

Th determination of bioactive compounds in mixtures such as EOs can provide valuable information on their potential application. Considering that the production of volatile constituents in plants is induced by various factors, their qualitative and quantitative analysis is a necessary step. Therefore, our first task was to evaluate the chemical composition of the essential oil further investigated in this study. The results presented in Table 1 expressed as a percentage of individual components are obtained by employing gas chromatography analysis. Additionally, results are sorted as percentage amounts of the different classes of compounds present in the tested sample, along with the experimentally determined RI. An overall of 59 components were successfully identified, which represents a total of 99.2%. Based on the presented results, the tested sample of TZEO is mainly characterized by a high abundance of monoterpene compounds (96.1%) among which oxygenated monoterpenes (61.8%) prevailed. Semi-quantitative analysis revealed monoterpene alcohol linalool as the major component of EO. With an abundance of 38%, linalool represents more than 60% of the total amount of identified oxygenated monoterpenes. Identified in considerably higher quantities were 4-terpineol (10.1%), *p*-cymene (6.9%), and γ-terpinene (6.5%), all belonging to the monoterpene class. Following this, notable amounts were also detected of β-myrcene (4.9%), α-pinene (3.8%), α-terpinene (3.2%), and limonene (2.7%) from the class of monoterpene hydrocarbons, 1,8-cineole (2.4%) from the monoterpene epoxide class, and borneol (3.5%) from the class of monoterpene alcohols. Another 49 compounds were quantified in the amounts ≤2%.

Bearing in mind that essential oils represent the volatile content of plant secondary metabolites, their production depends on many factors, such as genetic variations, harvesting time, climate factors, storage, etc. [39,40]. Moreover, part of the used plant, and the extraction method applied, have a significant influence on the extraction of these bioactive compounds. Previous reports indicate that thymol, carvacrol, and linalool are the two most common chemotypes of *T. zygis* [1]. Nonetheless, the literature provides a large amount of data regarding the thymol and carvacrol chemotypes, whereas data on the linalool chemotype are relatively scarce. In this regard, it is worth mentioning that the results presented in this study clearly indicate that the investigated TZEO belongs to the linalool chemotype. The obtained results, regarding the volatile composition of investigated TZEO, show a similarity with previously published findings. In this regard, a study conducted by Abdallah et al. [41] also indicates that this EO is a rich source of linalool (39.7%), followed by a significant concentration of 4-terpineol (11.7%) and γ-terpinene (7.6%). Cutillas et al. [42] showed similar results to those obtained in this study, indicating linalool and 4-terpineol as compounds presented in the highest amounts. Moreover, investigations performed by Rota et al. [43] revealed similar amounts of linalool and 4-terpineol, but with a different concentration of (*E*)-sabinene hydrate [43]. Additionally, in another study 1,8-cineole, *p*-cymene, and carvacrol were detected with high amounts [21]. All these results show valuable variations in the chemical profile within the same species. These variabilities confirm that the chemical profile highly depends on the origin of the plant, climate factors, genetic inheritance, abiotic factors, etc. [44].

The results presented in Table 1 designated the TZEO as a substantial source of biological activity compounds which prompted us to evaluate its biological features.

### 3.2. Antioxidant Activity

The antioxidant activity of TZEO has been assessed by employing ABTS radical cation and DPPH radical scavenging assays. Results have been obtained as IC_50_ values calculated graphically using the TZEO concentration versus the scavenging capacity (%). The IC_50_ value obtained for the DPPH assay was 1046.96 ± 2.59 μg/mL, whereas for the ABTS assay it was found to be 53.03 ± 1.34 μg/mL. Additionally, obtained results are expressed as the TEAC value. For the DPPH assay, the TEAC value was found at 0.004 ± 0.000, while for the ABTS assay, TEAC was found at 0.031 ± 0.001. Considering the presented results, it can be clearly concluded that TZEO has better neutralization activity of ABTS^•+^ compared to DPPH^•^. The study provided by Floegel et al. [45] indicated that the examination of plant foods containing hydrophilic, lipophilic, and high-pigmented antioxidant compounds shows the advantage of the ABTS assay compared to the DPPH assay. Additionally, a review article provided by da Silva et al. [46] demonstrated, again, that essential oils from the Lamiaceae family show better antioxidant activity against the ABTS^•+^ compared to DPPH^•^. 

Available literature data provide different information on the antioxidant potential of *T. zygis* essential oils, which can be explained by the variances of their chemical composition. Radi et al. [27] showed that *T. zygis* characterized by high amounts of carvacrol and *p*-cymene exerts antioxidant activity towards the DPPH radical with IC_50_ of the order of 6.13 ± 0.11 μg/mL. In another study, *T. zygis* with high proportions of thymol displayed TEAC values in order of 1061 ± 10 µmol TE/g EO for neutralization of ABTS radical cation, and 27.7 ± 0.3 µmol TE/g EO for neutralization of DPPH radical cation, while the same species containing high amounts of linalool showed activity towards ABTS^•+^ in the order of 10.6 ± 0.9 µmol TE/g EO and was not active towards DPPH^•^ [47]. These findings agree with the results obtained in this study.

### 3.3. In Vitro Antibacterial Activity Assessment

With the aim of evaluating the antimicrobial effects of TZEO, this study employs the two most used techniques. Therefore, first, TZEO was screened through the disc-diffusion susceptibility assay and the results are provided in Table 2. The presented results indicate that TZEO activity varies in response to the microorganism used in the study. Overall, from Table 2, it can be concluded that gram-negative (G^−^) bacteria were more sensitive to TZEO compared to gram-positive (G^+^) bacteria. The inhibition zones for G^−^ species vary from the most susceptible *E. aerogenes* (12.33 ± 0.58 mm), and slightly less sensitive *E. coli* (7.33 ± 0.58) mm, followed by *P. putida* (4.67 ± 0.58 mm), which was the most resistant. Considering G^+^ strains, observed inhibition zones were found to be in the range of 9.00 ± 1.00 mm (*L. monocytogenes*) to 5.67 ± 0.58 mm (*S. aureus*). For the *Candida* yeasts, TZEO showed the best inhibition power towards *C. albicans* and *C. krusei* (8.67 ± 0.58 mm), whereas *C. glabrata* (3.33 ± 0.58 mm) was found to be the most resistant of these strains. 

The presented results generally indicate good antimicrobial effects of the TZEO mixture, which prompted further antimicrobial investigations. In this regard, the MIC test was conducted, and the results are reported in Table 3. An analysis of these results showed the highest efficiency of TZEO towards *E. aerogenes* which expressed the lowest MIC 50 (0.147 ± 0.006 mg/mL) and MIC 90 (0.158 ± 0.024 mg/mL) values, while *L. monocytogenes* showed the highest MIC 50 (1.338 ± 0.123 mg/mL) and MIC 90 (1.456 ± 0.019) values. With an exception to the most resistant, *L. monocytogenes*, TZEO can be considered a potent inhibitor of tested bacterial and yeast species. 

Considering the biological profile of essential oils, it is worth remembering that variances in the results of different studies may be a consequence of the dissimilar chemical profile of each tested essential oil. In addition, high amounts of the main components, antagonistic, additive, or synergistic effects of the compounds presented in minor amounts cannot be neglected [48]. By reviewing the available literature, we concluded that the TZEO (linalool chemotype) investigations regarding antimicrobial testing are relatively limited. Previous reports indicate that the high concentrations of thymol, linalool, *p*-cymene, and carvacrol can be attributed to the antimicrobial effects of this EO [42]. The same study showed that two EOs obtained from *T. zygis* chem. linalool were effective against the *S. aureus* strain, with MIC values of 1.3 μL/mL for both. In the study conducted by Lagha et al. [49], TZEO (linalool) showed the most potent antibacterial action against *E. coli* isolates compared to the essential oils obtained from *O. majorana*, *R. officinalis*, *J. communis*, and *Z. officinale*. The antimicrobial tests of Rota et al. [43] showed that a higher concentration of linalool in EO obtained from two different *T. zygis* species is not necessarily a reason for the best antimicrobial effect. Moreover, lower concentrations of linalool in the tested samples showed better MIC and MBC values for different bacterial strains. These results clearly indicate the power of the synergist effect among major and minor volatiles presented in the EOs. Considering the TZEO (linalool) antifungal activity, Gonçalves et al. [21] showed that dermatophyte strains exhibited higher sensibility to the treatment of this EO compared with yeasts and filamentous fungi. In our study, it was also shown that Gram-negative bacteria are more susceptible to the activity of *T. zygis* essential oil compared to Gram-positive bacteria. However, contrary to our results, and based on previous findings, Gram-positive bacteria are known to be more susceptible to the effects of essential oils due to the structure of their cell walls [50]. 

### 3.4. In Situ Antibacterial Assessment

As previously mentioned, food-borne pathogens can induce major modifications in food safety and quality, which causes the increasing issue of foodborne diseases. As one of the solutions to this problem, recent research by the scientific community worldwide is based on exploiting plants and their products which represents a recognized source of antimicrobial agents of natural origin that are capable of controlling or preventing natural deterioration processes [1]. Among plant products, EOs have attracted significant attention, mainly because they are recognized as safe by the US FDA (Food and Drug Administration) and the EPA (Environmental Protection Agency). In addition, their application is supported by the fact that they are widely used in industry, with a particular focus on the food, cosmetic, and pharmaceutical industries [16]. 

In light of this, an in situ antimicrobial analysis was performed to clarify the effects of the TZEO vapor phase in food preservation. For this purpose, commercially available food models (banana and cucumber) were contaminated with the bacterial and yeast strains used for the in vitro antibacterial evaluation. 

Our results showed that TZEO (in all concentrations tested) had strong antibacterial efficacy against the growth of G^+^ and G^−^ bacteria, while towards yeast growth inhibition, this EO showed weak effects (Table 4). The best inhibition effect of TZEO against all microorganisms, on both dietary patterns, was found in a concentration of 500 µg/mL. The strongest antimicrobial activity was found against *E. coli*, followed by a large inhibition of *E. aerogenes* and *P. putida* in all concentrations, while the weakest effects were observed against *C. tropicalis* growing on the banana model. Considering the inhibition of microbial growth on the cucumber model, TZEO in the vapor phase expressed the best inhibition effects against *P. putida* and *E. aerogenes*, followed by the notable suppression of *E. coli* growth in all applied concentrations.

To our knowledge, there are no previously published results on the linalool TZEO chemotype vapor phase effects on microbial inhibition. However, literature data revealed that the use of 2 MIC values of *T. zygis* (thymol) EO in the chicken juice and lettuce model after two days of storage reduced *L. monocytogenes* counts (starting inoculum of 10^6^ CFU/mL) [26]. In the same study, authors concluded that after a 5 min immersion, the use of EO at the concentration of 0.2% (*v*/*v*) used as a sanitizer for fresh vegetables reduces *L. monocytogenes* and the natural microbiota to levels lower than those detectable by the method’s detection limit for iceberg lettuce and the spinach sample [26]. According to numerous studies, *T. zygis* essential oil has a wide range of antibacterial properties and may even enhance the effectiveness of some antimicrobial drugs. Its antibacterial efficacy against food spoilage and harmful microbes has been studied on several matrices, pointing to its potential use as a food preservative [1].

### 3.5. Antibiofilm Activity of TZEO against S. eneterica

Since TZEO has demonstrated efficiency in inhibiting planktonic bacterial cells, this study was further designed to examine the inhibitory potential of this essential oil in biofilm formation. By using MALDI-TOF MS Biotyper mass spectrometry, the impact of *T. zygis* essential oil against the biofilm-producing *S. enterica* was assessed and the results are presented in Figure 1. The formation of biofilm was observed on two different surfaces—plastic and stainless steel. Planktonic cells were used as a control to contrast the molecular alterations of the biofilm, and the spectra of the control groups evolved in the same manner. Control groups consisted of spectral data of planktonic cells and biofilm untreated with EO.

The obtained results show that right at the beginning of the experiment, the influence of TZEO on the biofilm formation of the experimental groups was observed. The effect of the treatment was reflected in the differences in the spectra of the two experimental groups (plastic and stainless steel) compared to the control planktonic spectra. Based on the recording of the evolution of the spectra, significant changes in the protein profile were observed, indicating a disturbance in the biofilm formation in experimental groups. The experimental group’s mass spectra continued to evolve differently than the control planktonic spectrum on day 5 (5SEP, 5SES, 5SEPC). Due to the destruction of biofilm in experimental groups, a change in the protein profile occurred. During the seventh day of the experiment, the same pattern was noticed. The mass spectrum of the plastic surface showed a permanent difference on the ninth day of the experiment, whereas the experimental group on stainless steel showed a return to the similarity with the control group. However, during the final two days of the experiment, we noticed a return to dissimilarity in the mass spectra of the experimental and control planktonic groups on both surfaces. The general conclusion of the presented results is that the TZEO impacts the disruption of *S. enterica* biofilm homeostasis, leading to its suppression on both experimental surfaces from the early stage of the biofilm formation.

Additionally, the dendrogram was created based on the presented results (Figure 2). The smallest MSP distances were found between planktonic cells and controls. Over the duration of the trial, increasing MSP distances were observed for the experimental group. For the third day of the experiment, the smallest MSP distance was observed for the experimental groups. The longest MSP distance of the experimental group can be seen dominating the plastic surface on the fourteenth day. However, on day 9, the MSP distance for the experimental group acquired from the stainless-steel surface was shortening. Based on the presented results, our research established the inhibitory and destructive effects of *T. zygis* essential oil on *S. enterica* forming biofilms on plastic and stainless-steel surfaces.

Previously conducted studies have shown that *T. zygis* EO (thymol) significantly reduced the biofilm formation (inhibition from 16.85 to 89.86%) and motility (halos between 6.66 and 10.98 mm) of *L. monocytogenes* without causing cross-resistance to antibiotics [26]. The findings of Abdallah et al. [41] demonstrated the ability of *T. zygis* EO of the linalool chemotype to inhibit biofilm formation of 18 MRSA (Methicillin-Resistant *Staphylococcus aureus*) isolates in a range of 11.67 to 91.48% while showing biofilm eradication on 12 isolates ranging from 12.65 to 94.39%. Another study on TZEO of the linalool chemotype demonstrated its inhibitory power against 14 different *E. coli* UTI isolates, with inhibition percentages ranging from 17.81% to 85.81% [49]. The authors of this study raised the possibility that the addition of EOs prior to biofilm formation could help eliminate planktonic *E. coli* cells and modify the abiotic surface in a way that makes it less prone to cell adhesion. In addition, several reports have shown that the biofilms could be effectively removed by the application of EOs [51,52,53].

### 3.6. Determination of Cell Viability (MTT Assay)

With the aim of further elucidating the biological properties of TZEO of the linalool chemotype, we examined its antiproliferative effects. The experiments performed on normal human fibroblast cell line MRC-5, human breast cancer MDA-MB-231, colon cancer HCT-116, and chronic myelogenous leukemia K562 cell line were used to determine the effects of TZEO on the viability and redox status of cells. All cell lines were exposed to treatment with TZEO for 24 h (short-term) and 72 h (long-term). 

The selectivity of the tested essential oil was performed by employing the MTT test on non-transformed fibroblasts (MRC-5). The effect of the TZEO on the proliferation potential of MRC-5 during 24 h and 72 h treatment is shown in Figure 3. The presented results show that the proliferation level of cultivated MRC-5 cells was higher than 83% (drop of 17%) compared to non-treated cells, demonstrating favorable biocompatibility towards healthy tissue, which qualifies this EO as suitable for further evaluation.

Accordingly, TZEO effects on the cell viability of a panel of three cancer cell lines (MDA-MB-231, HCT-116, and K562) were assessed. The results of cell viability after 24 and 72 h of exposure to the treatments are presented in Figure 4. Overall, the obtained results indicate that all applied concentrations of EOs after 24 h caused a decrease in cell viability of all exposed types of cells, especially in concentrations 100 and 200 µg/mL. After 72 h of treatments, the viability of subjected cells was significantly reduced compared to the treatment after 24 h of exposure, indicating a time- and dose-dependent decrease.

By comparing the obtained results regarding the proliferative effect of TZEO on the tested cell lines, it can be concluded that this essential oil exerted the strongest effects on the inhibition of the viability of MDA-MB-231, especially after long-term treatment and in the highest applied concentrations (100 µg/mL and 200 µg/mL). Moreover, TZEO exerted significantly stronger antiproliferative effects in MDA-MB-231 (57% after 72 h) cells than in MRC-5 cells, making a drop of 43% in cell viability compared to non-treated cells. The effects of TZEO on HCT 116 and K562 cell lines also demonstrated strong antiproliferative potential, after long-term treatment, with inhibition rates of 67% and 60%, respectively. Compared to the effects of TZEO on MRC-5 cells, the observed drop on HCT 116 cells was 33%, while for K562 it was 40%. Additionally, presented results suggest that this oil exhibits desirable biocompatibility for potential further usage against cancer cells. 

Considering the literature regarding the antiproliferative activity of TZEO linalool chemotype, data are scarce. To the best of our knowledge, there are no previous reports on the anti-proliferative potential of this TZEO. However, in one study authors showed that the TZEO of carvacrol type showed no cytotoxic effect after 24 h of treatment [21]. Additionally, Delgado-Adámez et al. [23] showed that TZEO obtained from flowers and fruits of the plant (both thymol chemotypes) possess cytotoxic activity in vitro towards HeLa (adherent cells) and U937 (free-floating cells). The authors indicated that TZEO obtained from flowers and fruit induces the strongest cytotoxicity on (adherent) HeLa cells, with cell viability reduced to 82% and 75%, respectively [23].

### 3.7. The Effects of TZEO on Redox Status Parameters in Tumor Cells

It is well-known that one of the various indicators of oxidative metabolism in tumor cells is oxidative stress. As a key factor in the development of tumors, oxygen radicals are important regulators of numerous signaling pathways in cells. With this in mind, there is a widespread belief that some drugs produced from a biological source could increase the impacts of cytotoxic regimes. One explanation is that those therapeutics can modify redox homeostasis and improve the response rate of tumors [54]. In addition to oxygen radicals, nitric oxide represents a valuable signal molecule with antitumor and protumor properties. The presence of higher doses of superoxide anion radical decreases NO levels, which are strongly determined by the redox homeostasis of the cell. Changes in its concentrations can have an important impact on the cell cycle, proliferation, and metabolism, potentially having an anticancer effect [55]. 

Therefore, in order to obtain more complete information on the antiproliferative activity of TZEO, its effects on redox status parameters were observed on the same panel of three cancer cell lines. Consequently, the production of the oxidative stress indicators O_2_^•–^ (superoxide anion radical) and nitrites (NO_2_^−^) was determined. 

The effect of short-term and long-term exposure of human breast cancer MDA-MB-231, colon cancer HCT-116, and chronic myelogenous leukemia, K562 cell lines to different concentrations of TZEO treatment on redox status parameters (O_2_^•–^ and NO) were monitored, and the obtained results are shown in Figure 5 and Figure 6. The values of the obtained results are expressed in nmol/mL in 10^5^ cells for superoxide anion radical and µmol/mL in 10^5^ cells for nitrites. The results show that the production of O_2_^•–^ was decreased compared to control after a short amount of exposure to all concentrations in all tested cell lines. The same trend was continued after a longer time exposure of cells to the TZEO. However, TZEO treatment induced the most intensive reduction of this parameter in all tested cells in the highest concentration applied (200 µg/mL). 

Nitric oxide (NO) is an important signaling molecule in numerous physiological and pathological conditions. Regarding NO production, the obtained results show that after 24 h of treatment with TZEO, the concentration of NO_2_^−^ (as an indicator of NO) was increased after treatment with all concentrations compared to control cells. The most intensive increase in NO concentration was induced by the concentration of 200 µg/mL after long-term treatment. TZEO treatments showed a significant increase in the production of nitrite in all cell lines compared to the control. Since NO is a powerful signaling molecule, the increase could affect various metabolic pathways. 

The obtained results suggest that the antioxidative effects of TZEO recorded in the study through the decrease in O_2_^•−^ levels could be one of the main mechanisms of cancer cell growth restriction capacity of the tested oil. Since the levels of NO production are high in all three cell lines, in all applied concentrations at both time treatments we could suggest that the correlation with the decrease in O_2_^•−^ concentrations is significant and that the high bioavailability of NO is due to a low oxidative environment. The obtained data indicate that the stimulation of NO production and/or bioavailability significantly contributes to the recorded antitumor activity of the tested EO. The obtained data suggest that TZEO could be a cost-effective candidate for future testing with the aim of developing novel antitumor therapies. 

## 4. Conclusions

The results presented in this study indicate that TZEO of the linalool chemotype possesses good biological activity. The performed GC/MS analysis characterized this sample as a source of biologically active linalool, 4-terpineol, *p*-cymene, and γ-terpinene. Antioxidant assays revealed TZEO as a better agent in neutralizing ABTS^•+^ compared to DPPH^•^. Antimicrobial tests revealed generally good activity of TZEO, and Gram-negative bacterial strains are the most susceptible to treatment with this EO. The vapor phase antimicrobial examination of this EO showed generally good effects in microbial inhibition on both food models (banana and cucumber), with the most pronounced activity at the highest applied concentration. Nonetheless, our results indicate that TZEO had the best effects in inhibiting the growth of Gram-negative bacteria *E. aerogenes*, *E. coli*, and *P. putida* in both dietary models. Based on the results on the effects of TZEO on the preventing biofilm formation of *S. enterica* growing on plastic and stainless-steel surfaces, a general conclusion is that this EO affects the disruption of *S. enterica* biofilm homeostasis, leading to its suppression on both experimental surfaces from the early stage of the biofilm formation. Our further investigations demonstrated that TZEO exerted a mild effect on the viability of the tested normal MRC-5 cell line. In contrast, TZEO induced a strong time- and dose-dependent decrease in the viability of all three cancer cell lines (MDA-MB-231, HTC-116, and K562). Compared to the effect on normal MRC-5 cell line, TZEO exerted a multifold stronger antiproliferative effect on cancer cells with the dropping of cell viability of 2.53, 1.94, and 2.33-fold for MDA-MB-231, HCT-116, and K562, respectively. Further, we performed the examination of redox status parameters on the tested cells after TZEO treatment. These results suggested that TZEO causes a decrease in O_2_^•−^ concentrations and stimulates NO production. 

Based on the presented findings, the first conclusion of our study is that tested TZEO could be an effective agent in extending the shelf-life and safety of food products, as well as a promising antibiofilm agent in combat against *S. enterica*. Additionally, mild toxicity toward normal MRC-5 cell line qualifies this EO as suitable for further evaluation. Moreover, the obtained results on cell viability and redox status parameters of cancer cell lines indicate that TZEO could be a suitable candidate for future tests in the development of novel anticancer therapies.

## Figures and Tables

**Figure 1 plants-12-03920-f001:**
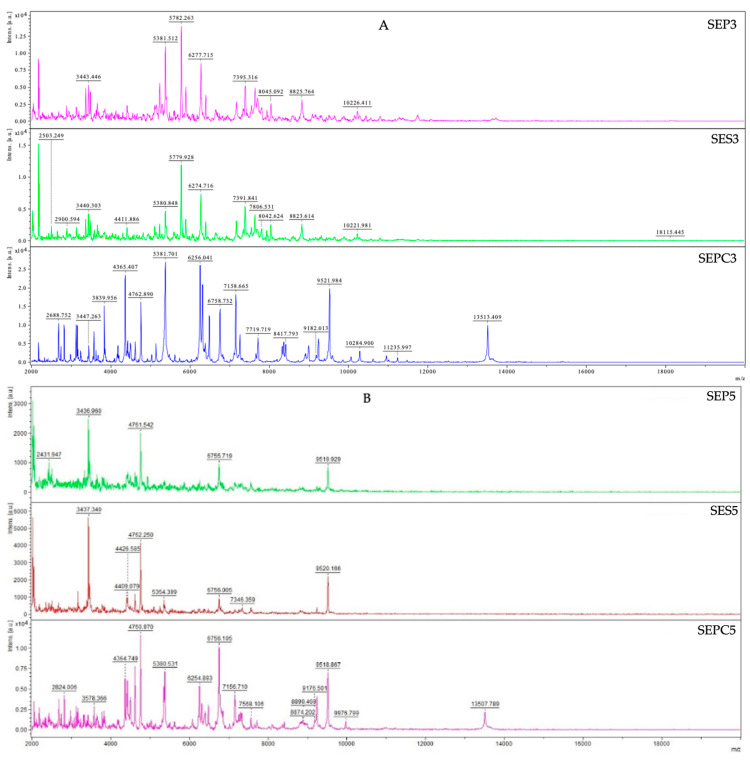
MALDI-TOF mass spectra of *S. enterica* during the development of the biofilm: SE—*Salmonella enterica*, PC—planktonic cell, P—plastic, S—stainless steel; (**A**)—3rd day; (**B**)—5th day; (**C**)—7th day; (**D**)—9th day; (**E**)—12th day; (**F**)—14th day.

**Figure 2 plants-12-03920-f002:**
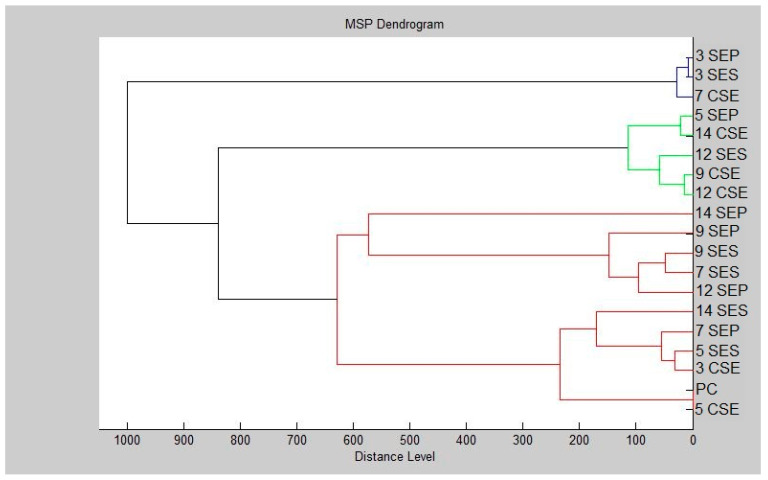
Dendrogram of *S. enterica* biofilm progress after TZEO exposition: SE—*Salmonella enterica*, PC—planktonic cell, P—plastic, S—stainless steel, and C—untreated biofilm.

**Figure 3 plants-12-03920-f003:**
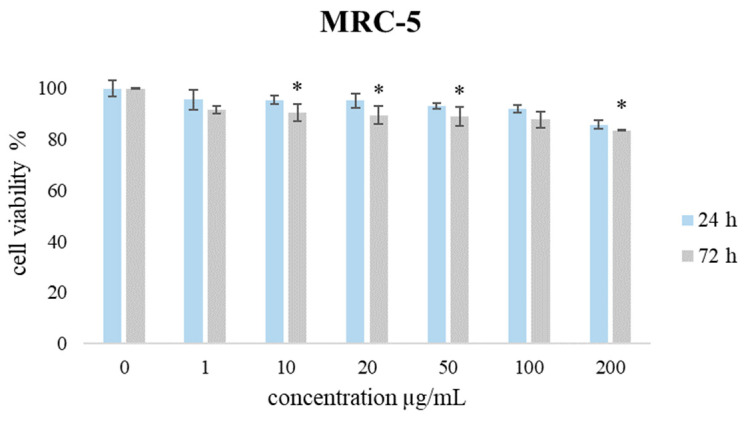
The effects of TZEO on the viability of MRC-5 cells after 24 and 72 h of treatment. Results are presented as the mean of three independent experiments ± standard error: * *p* < 0.05 relative to control.

**Figure 4 plants-12-03920-f004:**
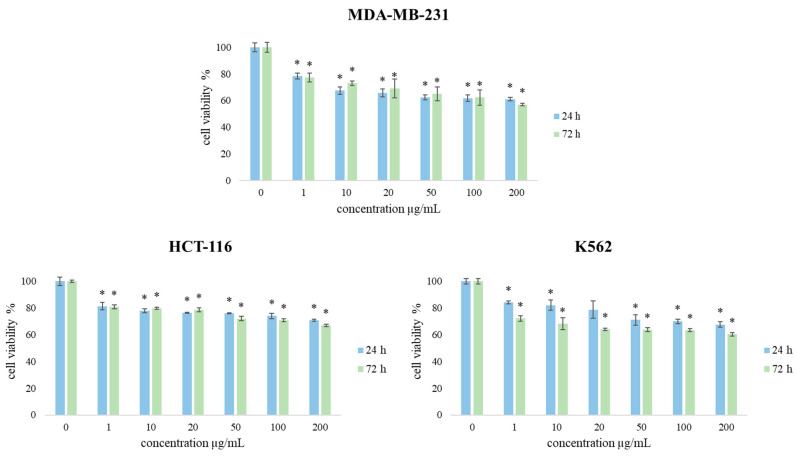
The effects of TZEO on the viability of MDA-MB-231, HCT-116, and K562 cells after 24 and 72 h of treatment. Results are presented as the mean of three independent experiments ± standard error: * *p* < 0.05 relative to control.

**Figure 5 plants-12-03920-f005:**
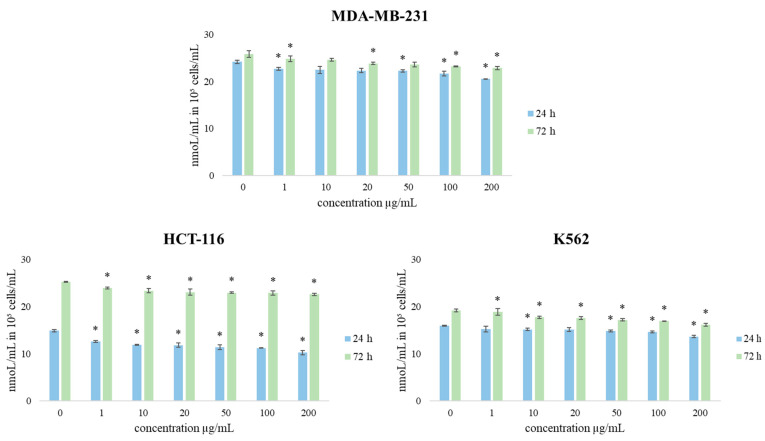
The effects of TZEO on the concentration of O_2_^•–^ in MDA-MB-231, HCT-116, and K562 cells after 24 and 72 h of treatment. Results are presented as the mean of three independent experiments ± standard error: * *p* < 0.05 relative to control.

**Figure 6 plants-12-03920-f006:**
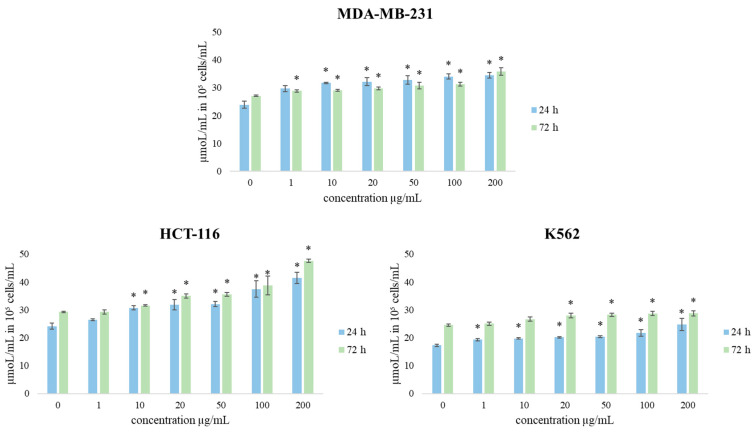
The effects of TZEO on the concentration of NO_2_^−^ in MDA-MB-231, HCT-116, and K562 cells after 24 and 72 h of treatment. Results are presented as the mean of three independent experiments ± standard error: * *p* < 0.05 relative to control.

**Table 1 plants-12-03920-t001:** Chemical composition of *T. zygis* essential oil.

No	RI (Lit.)	RI (Calc.) ^a^	Rt ^b^	Compound ^c^	%
				Monoterpenes	96.1
				*monoterpene hydrocarbons*	34.3
1	926	924	7.22	Tricyclene	0.1
2	930	927	7.32	α-thujene	0.4
3	939	935	7.60	α-pinene	3.8
4	954	951	8.16	Camphene	1.7
5	967	954	8.28	Verbenene	tr ^d^
6	975	972	8.98	Sabinene	0.8
7	979	976	9.17	β-pinene	0.9
8	990	987	9.62	β-myrcene	4.9
9	1000	999	10.18	*m*-mentha-1(7),8-diene	0.1
10	1002	1001	10.28	α-phellandrene	0.4
11	1011	1003	10.36	δ-3-carene	tr
12	1017	1012	10.75	α-terpinene	3.2
13	1024	1021	11.13	*p*-cymene	6.9
14	1029	1026	11.36	Limonene	2.7
15	1029	1027	11.39	β-phellandrene	0.3
16	1050	1042	12.15	(*E*)-β-ocimene	0.3
17	1059	1056	12.84	γ-terpinene	6.5
18	1088	1084	14.40	Terpinolene	1.3
				*oxygenated monoterpenes*	61.8
				*monoterpene epoxides*	2.9
19	1031	1028	11.47	1,8-cineole	2.4
20	1086	1087	14.53	trans-linalool oxide (furanoid)	0.3
21	1244	1241	21.51	methyl ether carvacrol	0.2
				*monoterpene alcohols*	56.4
22	1070	1069	13.55	cis-sabinene hydrate	1.0
23	1096	1108	15.68	Linalool	38.0
24	1121	1130	16.65	cis-*p*-menth-2-en-1-ol	0.2
25	1133	1149	17.53	1-terpineol	0.2
26	1160	1168	18.46	Isoborneol	tr
27	1169	1177	18.91	Borneol	3.5
28	1177	1186	19.33	4-terpineol	10.1
29	1182	1189	19.51	*p*-cymen-8-ol	0.1
30	1188	1196	19.87	α-terpineol	2.0
31	1216	1220	20.77	trans-carveol	tr
32	1229	1226	20.99	Nerol	tr
33	1290	1291	23.35	Thymol	1.1
34	1299	1297	23.61	Carvacrol	0.2
				*monoterpene ketones*	1.9
35	1146	1153	17.72	Camphor	1.5
36	1192	1197	19.96	cis-dihydro carvone	0.2
37	1205	1206	20.31	Verbenone	0.2
				*monoterpene aldehydes*	tr
38	1267	1270	22.57	geranial	tr
				*monoterpene esters*	0.6
39	1285	1285	23.13	bornyl acetate	0.2
40	1349	1348	25.12	α-terpinyl acetate	0.1
41	1361	1360	25.48	neryl acetate	tr
42	1381	1379	26.07	geranyl acetate	0.3
				sesquiterpenes	2.6
				*sesquiterpene hydrocarbons*	2.3
43	1388	1384	26.23	β-bourbonene	tr
44	1390	1389	26.39	β-elemene	tr
45	1419	1421	27.29	trans-caryophyllene	2.0
46	1434	1434	27.63	α-trans-bergamotene	tr
47	1441	1440	27.79	aromadendrene	tr
48	1454	1457	28.26	α-humulene	tr
49	1484	1476	28.77	α-amorphene	tr
50	1496	1491	29.20	viridiflorene	tr
51	1500	1496	29.33	bicyclogermacrene	0.1
52	1500	1498	29.39	α-muurolene	tr
53	1505	1507	29.62	β-bisabolene	tr
54	1513	1519	29.89	γ-cadinene	0.2
				*oxygenated sesquiterpenes*	0.6
				*sesquiterpene alcohols*	0.1
55	1578	1579	31.38	spathulenol	0.1
				*sesquiterpene epoxides*	0.5
56	1582	1585	31.52	caryophyllene oxide	0.2
				non-terpenic compounds	0.5
				*ketones*	0.1
57	983	983	9.45	3-octanon	0.1
				*alcohols*	0.4
58	991	993	9.90	3-octanol	tr
59	1110	1109	15.72	hotrienol	0.4
				total	99.2

^a^ Values of retention indices on HP-5MS column; ^b^ Retention time on HP-5MS column; ^c^ Identified compounds; ^d^ tr-compounds identified in amounts less than 0.1%.

**Table 2 plants-12-03920-t002:** Antimicrobial activity of TZEO obtained by disc diffusion assay displayed in mm.

Microorganism	Inhibition Zone ^a^	ATB ^b^
Gram-positive bacteria		
* Listeria monocytogenes *	9.00 ± 1.00	23.33 ± 0.08
* Micrococcus luteus *	6.67 ± 0.58	25.67 ± 0.03
* Staphylococcus aureus *	5.67 ± 0.58	27.33 ± 0.04
Gram-negative bacteria		
* Enterobacter aerogenes *	12.33 ± 0.58	21.67 ± 0.03
* Escherichia coli *	7.33 ± 0.58	24.67 ± 0.02
* Pseudomonas putida *	4.67 ± 0.58	27.33 ± 0.04
Yeasts		
* Candida albicans *	8.67 ± 0.58	25.67 ± 0.03
* Candida glabrata *	3.33 ± 0.58	28.67 ± 0.03
* Candida krusei *	8.67 ± 0.58	27.67 ± 0.04
* Candida tropicalis *	6.67 ± 0.58	24.67 ± 0.02

^a^ Inhibition zones are presented in mm. ^b^ Antibiotics (ATB) used as a control are the following: cefoxitin for G^−^ bacteria, gentamicin for G^+^ bacteria, and fluconazole for yeasts.

**Table 3 plants-12-03920-t003:** Minimal inhibitory concentrations of TZEO.

Microorganism	MIC 50	MIC 90
Gram-positive bacteria		
*Listeria monocytogenes*	1.338 ± 0.123	1.456 ± 0.019
*Micrococcus luteus*	0.184 ± 0.016	0.234 ± 0.023
*Staphylococcus aureus*	0.236 ± 0.018	0.267 ± 0.014
Gram-negative bacteria		
*Enterobacter aerogenes*	0.147 ± 0.006	0.158 ± 0.024
*Escherichia coli*	0.228 ± 0.010	0.248 ± 0.021
*Pseudomonas putida*	0.217 ± 0.012	0.235 ± 0.012
Yeasts		
*Candida albicans*	0.370 ± 0.027	0.412 ± 0.015
*Candida glabrata*	0.273 ± 0.017	0.304 ± 0.032
*Candida krusei*	0.286 ± 0.021	0.322 ± 0.015
*Candida tropicalis*	0.226 ± 0.008	0.247 ± 0.016

MIC 50 and MIC 90 values are presented in mg/mL.

**Table 4 plants-12-03920-t004:** In situ analysis of the antibacterial activity of the vapor phase of TZEO on banana and cucumber.

Food Model	Microorganisms	Inhibition of Microbial Growth (%)			
		Concentration of EO in μg/L			
		62.5	125	250	500
Banana					
G^+^	*Listeria monocytogenes*	14.23 ± 5.78	28.55 ± 3.45	43.37 ± 4.29	67.35 ± 5.43
	*Micrococcus luteus*	15.78 ± 5.22	28.56 ± 3.84	46.75 ± 6.45	78.46 ± 4.56
	*Staphylococcus aureus*	18.48 ± 5.32	27.37 ± 3.54	48.34 ± 5.36	79.34 ± 4.37
G^−^	* Enterobacter aerogenes *	25.63 ± 5.34	45.26 ± 3.28	75.43 ± 4.63	93.47 ± 3.43
	* Escherichia coli *	32.46 ± 4.34	65.43 ± 5.34	78.12 ± 4.34	95.64 ± 4.23
	* Pseudomonas putida *	25.26 ± 5.43	49.32 ± 5.43	76.37 ± 6.23	91.24 ± 4.41
Yeasts	* Candida albicans *	8.54 ± 3.42	17.34 ± 2.43	34.58 ± 5.32	45.76 ± 4.12
	* Candida glabrata *	7.56 ± 2.76	16.34 ± 4.24	35.84 ± 4.34	44.24 ± 4.18
	* Candida krusei *	10.34 ± 5.23	26.51 ± 5.48	43.21 ± 6.79	83.87 ± 4.76
	* Candida tropicalis *	6.34 ± 5.54	16.34 ± 4.46	27.24 ± 4.27	37.34 ± 3.54
Cucumber					
G^+^	*Listeria monocytogenes*	17.65 ± 3.34	23.56 ± 3.54	44.26 ± 3.44	58.45 ± 4.65
	*Micrococcus luteus*	22.48 ± 3.34	31.36 ± 3.46	54.39 ± 4.26	76.34 ± 4.34
	*Staphylococcus aureus*	23.42 ± 5.63	42.53 ± 5.62	63.47 ± 5.33	82.53 ± 3.42
G^−^	* Enterobacter aerogenes *	34.67 ± 4.23	57.34 ± 3.28	71.58 ± 4.34	93.24 ± 4.61
	* Escherichia coli *	33.56 ± 5.34	52.65 ± 5.28	65.74 ± 4.67	83.87 ± 4.76
	* Pseudomonas putida *	36.32 ± 5.54	62.34 ± 3.26	75.44 ± 4.25	95.47 ± 4.73
Yeasts	* Candida albicans *	14.34 ± 3.36	24.37 ± 2.87	34.28 ± 5.33	45.34 ± 3.47
	* Candida glabrata *	15.67 ± 4.36	27.74 ± 4.86	35.43 ± 4.34	44.46 ± 4.36
	* Candida krusei *	13.36 ± 3.24	22.59 ± 2.86	35.45 ± 4.75	43.17 ± 4.16
	* Candida tropicalis *	16.34 ± 5.28	26.48 ± 4.28	37.44 ± 6.22	48.74 ± 4.72

## Data Availability

Data are contained within the article.

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
