# Peer review of "Thymus zygis, Valuable Antimicrobial (In Vitro and In Situ) and Antibiofilm Agent with Potential Antiproliferative Effects"

_plants, 2023, doi:10.3390/plants12233920_

Round 1

Reviewer 1 Report

Comments and Suggestions for Authors

The reviewed article presupposes a complex, well-organized research. The topic of the research presents a great interest.

The bibliography includes recent research, which shows an increased interest in the effects of Thymus zygis.

I agree to the publication of the article with a comment. In the abstract, lines 32-34 are repeated. It would be interesting to note if the tested product has a standardized composition

Author Response

Reviewer #1

The reviewed article presupposes a complex, well-organized research. The topic of the research presents a great interest.

The bibliography includes recent research, which shows an increased interest in the effects of Thymus zygis.

I agree to the publication of the article with a comment. In the abstract, lines 32-34 are repeated. It would be interesting to note if the tested product has a standardized composition.

Response:

The Authors are very grateful to the Reviewer for the valuable comments. According to the comments we have deleted the redundant sentence in lines 34-36.

Reviewer 2 Report

Comments and Suggestions for Authors

1. Table 1.       Retention times should be inserted

2. Table 2         Please clarify the meaning of   ATB

3. The antimicrobial assaying should include some reference antibiotics as standard drugs for comprising 

4. Table 4       The names of all microorganisms should be written as full names

5. The abstract should be rewritten with some results that exhibited the significance of the oil

6. Figure 3-6     is very unclear and should be re-constructed

7. The data sources of the" Figure 2. Dendrogram of S. enterica biofilm progress after TZEO exposition" should be inserted as supporting data 

Comments on the Quality of English Language

minor revisions needed

Author Response

Reviewer #2

The Authors are very grateful to the Reviewer for the valuable comments. We would like to thank the Reviewer for the time devoted for constructive and important comments to improve our paper.

Point 1. Table 1.       Retention times should be inserted.

Response:

We have added retention times in Table 1.

Point 2. Table 2.   Please clarify the meaning of   ATB

Response:

We have clarified the meaning of ATB in the footer of Table 2.

Point 3. The antimicrobial assaying should include some reference antibiotics as standard drugs for comprising.

Response:

We have included the reference antibiotics for comprising in the disc-diffusion assay (ABT column).

Point 4. Table 4       The names of all microorganisms should be written as full names.

Response:

Dear reviewer, we have added the full names of microorganisms in Table 4.

Point 5. The abstract should be rewritten with some results that exhibited the significance of the oil.

Response:

We have rewritten the abstract according to the suggestions. Please see lines 24-46.

Point 6. Figure 3-6     is very unclear and should be re-constructed.

Response:

We have reconstructed Figures 3-6, and hope they are clearer now.

Point 7. The data sources of the" Figure 2. Dendrogram of S. enterica biofilm progress after TZEO exposition" should be inserted as supporting data 

Response:

Dear reviewer, the authors are grateful for the suggestion. However, our opinion is that the dendrogram is necessary in order to additionally confirm the presented results.

Reviewer 3 Report

Comments and Suggestions for Authors

The paper entitled „ Thymus zygis, valuable antimicrobial (in vitro and in situ) and antibiofilm agent with potential antiproliferative effects“ by Vukić et al. is very well written, so I recommend to be published in Plants after a minor revision of the manuscript:

1.       In lines 32-34, The sentence "Furthermore, a test on the..." is written once again in lines 34-36!

2.       In line 238,  Convert 12 000 rpm to rcf (xg)

3.       In line 283,  The concentration you wrote in parentheses is redundant. You can write 1% w/v instead.

Author Response

Reviewer #3

 The paper entitled „ Thymus zygis, valuable antimicrobial (in vitro and in situ) and antibiofilm agent with potential antiproliferative effects“ by Vukić et al. is very well written, so I recommend to be published in Plants after a minor revision of the manuscript:

Response: The Authors would like to thank the Reviewer for the time devoted for constructive and important comments to improve our paper.

Point 1.       In lines 32-34, The sentence "Furthermore, a test on the..." is written once again in lines 34-36!

Response: We have deleted the redundant sentence in lines 34-36.

Point 2.       In line 238,  Convert 12 000 rpm to rcf (xg)

Response: We have converted rpm units in to 13 400 x g.

Point 3.       In line 283,  The concentration you wrote in parentheses is redundant. You can write 1% w/v instead.

Response: We have changed the concentration units.

Reviewer 4 Report

Comments and Suggestions for Authors

In manuscript plants-2681020, the authors have presented the essential oil composition of Thymus zygis (linalool chemotype) as well as radical-scavenging, antibacterial, antifungal, antibiofilm, and tumor-cell cytotoxic activities. There are several points that should be addressed before publication can be recommended.

Abstract: “…minimal inhibitory concentration assays revealed TZEO as a potent antimicrobial agent…” The potency as an antimicrobial agent should be discussed with respect to previously published criteria for potency (Section 3.3). See, for example, Sartoratto et al., Brazilian Journal of Microbiology, 2004, 35, 275-280. These workers have suggested that essential oils with MIC values of 50-500 μg/mL are strong, 600-1500 μg/mL are moderate, and > 1500 μg/mL are weak. On the other hand, Cos et al. (Journal of Ethnopharmacology, 2006, 106, 290-302) suggest that “Relevant and selective activity relates to IC50 values below 100 μg/mL…” for plant extracts.

Table 1, entry 13: Please double-check; this may be p-cymene. See: Romanenko & Tkachev, Chemistry of Natural Compounds, 2006, 42(6), 699-701 (p-Cymene, RI = 1024; m-Cymene, RI = 1022; o-Cymene, RI = 1039). Since there are relatively large quantities of menthane monoterpenoids in TZEO, p-cymene seems more likely from a biosynthetic perspective and would agree with previously published compositions of TZEO. Table 1, entry 24: cis-p-menth-2-en-1-ol [“p” for “para”]. Table 1, entry 33: geranial is an aldehyde, not an alcohol.

Section 3.2: The discussion should be expanded. How do the DPPH and ABTS assay results compare with the values for other essential oils (i.e., how effective is TZEO as a radical-scavenging agent)? See, for example: da Silva et al., Evidence-Based Complementary and Alternative Medicine, 2021, Article ID 6748052. How does the antioxidant activity of TZEO compare with the major components? See: Ruberto & Baratta, Food Chemistry, 2000, 69, 167-174; Sharopov et al., Natural Product Communications, 2015, 10, 153-156.

Section 3.4, Table 4: Given these data, it should be possible to determine IC50 values, which may allow for better comparison.

Conclusions, lines 622-624: It is not clear how “multifold stronger” cytotoxic effect for MDA-MB-231 cells vs. MRC-5 cells can be concluded. From Figures 3 and 4, all cell lines show IC50 values > 200 ug/mL (i.e., not active according to the criteria of Cos et al.). In order to provide comparison with the normal cell line, IC50 values should be determined and a selectivity index determined to indicate relative safety of the essential oil. The discussion in Section 3.6 should be expanded accordingly.

Some minor editorial corrections:

·         Lines 34-36: This sentence is repeated and should be deleted.

·         Line 88: medicine [correct the spelling]

·         Line 99: Gram-positive and Gram-negative [Gram is a proper noun and should be capitalized]

Author Response

Reviewer #4

In manuscript plants-2681020, the authors have presented the essential oil composition of Thymus zygis (linalool chemotype) as well as radical-scavenging, antibacterial, antifungal, antibiofilm, and tumor-cell cytotoxic activities. There are several points that should be addressed before publication can be recommended.

Response:

The Authors are very grateful to the Reviewer for their valuable comments. We want to thank the Reviewer for the time devoted to point out constructive and important comments to improve our paper.

Point 1. Abstract: “…minimal inhibitory concentration assays revealed TZEO as a potent antimicrobial agent…” The potency as an antimicrobial agent should be discussed with respect to previously published criteria for potency (Section 3.3). See, for example, Sartoratto et al., Brazilian Journal of microbiology200435, 275-280. These workers have suggested that essential oils with MIC values of 50-500 μg/mL are strong, 600-1500 μg/mL are moderate, and > 1500 μg/mL are weak. On the other hand, Cos et al. (Journal of Ethnopharmacology2006106, 290-302) suggest that “Relevant and selective activity relates to IC50 values below 100 μg/mL…” for plant extracts.

Response:

The authors are grateful for the reviewer's constructive comment. However, the results obtained in this study were interpreted as MIC50 and MIC90 (minimal concentration able to inhibit 50% and 90% of bacterial growth) values, not as MIC (minimal inhibitory concentration), considering we couldn’t apply the suggested criteria for potency.

Initially, we also considered the approach to discuss our results regarding some previously published potency criteria, in order to more clearly interpret the antimicrobial potency of TZEO. However, a literature search exposed that such potency criteria are not consistent and generally accepted, since different publications are suggesting different MIC values to be considered as relevant. Here, besides the suggested criteria of Sartoratto et al. and Cos et al., some authors suggest that MIC values below 2 mg/mL are relevant, whereas other researchers indicate MIC values <1 mg/mL, or even <250 μg/mL as noteworthy (S.F. Van Vuuren, Y. Docrat, G.P.P. Kamatou, A.M. Viljoen, Essential oil composition and antimicrobial interactions of understudied tea tree species, South African Journal of Botany, Volume 92, 2014, Pages 7-14, ISSN 0254-6299, https://doi.org/10.1016/j.sajb.2014.01.005; Ané Orchard, Sandy van Vuuren, "Commercial Essential Oils as Potential Antimicrobials to Treat Skin Diseases", Evidence-Based Complementary and Alternative Medicine, vol. 2017, Article ID 4517971, 92 pages, 2017. https://doi.org/10.1155/2017/4517971 ). As these authors indicate, such activity classifications are presented as recommendations, based on the frequency of certain MIC values. Bearing in mind that essential oils are complex mixtures of multiple individual components, whose bioactivity properties are dependent on the presence and amount of both major and minor constituents, we found it inappropriate to discuss our results in relation to previously published potency criteria. Generally, essential oils are diverse in numerous ways, including bioactivity potency, especially considering the potential synergistic effect of minor components that enhance the activity. Also, the MIC values in different publications are obtained by different methods or interpreted by different concentration units, which makes the comparison of the results difficult to implement. Regarding all previously mentioned, we found ourselves discouraged from interpreting the antimicrobial activity with respect to previously published criteria for potency.

In addition, we are providing some of the papers discussing the antibacterial activity of different essential oils published recently which also don’t include the potency criteria:

  1. Ghavam, M., Manca, M.L., Manconi, M. et al. Chemical composition and antimicrobial activity of essential oils obtained from leaves and flowers of Salvia hydrangea DC. ex Benth.. Sci Rep 10, 15647 (2020). https://doi.org/10.1038/s41598-020-73193-y
  2. Trong Le, N.; Viet Ho, D.; Quoc Doan, T.; Tuan Le, A.; Raal, A.; Usai, D.; Madeddu, S.; Marchetti, M.; Usai, M.; Rappelli, P.; et al. In Vitro Antimicrobial Activity of Essential Oil Extracted from Leaves of Leoheo domatiophorus Chaowasku, D.T. Ngo and H.T. Le in Vietnam. Plants 2020, 9, 453. https://doi.org/10.3390/plants9040453
  3. Ghaffari, T.; Kafil, H.S.; Asnaashari, S.; Farajnia, S.; Delazar, A.; Baek, S.C.; Hamishehkar, H.; Kim, K.H. Chemical Composition and Antimicrobial Activity of Essential Oils from the Aerial Parts of Pinus eldarica Grown in Northwestern Iran. Molecules 2019, 24, 3203. https://doi.org/10.3390/molecules24173203
  4. Teneva, Desislava, Denkova-Kostova, Rositsa, Goranov, Bogdan, Hristova-Ivanova, Yana, Slavchev, Aleksandar, Denkova, Zapryana and Kostov, Georgi. "Chemical composition, antioxidant activity and antimicrobial activity of essential oil from Citrus aurantium L zest against some pathogenic microorganisms" Zeitschrift für Naturforschung C, 2019, vol. 74, no. 5-6, 2019, pp. 105-111. https://doi.org/10.1515/znc-2018-00624
  5. EL Moussaoui, A.; Bourhia, M.; Jawhari, F.Z.; Salamatullah, A.M.; Ullah, R.; Bari, A.; Majid Mahmood, H.; Sohaib, M.; Serhii, B.; Rozhenko, A.; et al. Chemical Profiling, Antioxidant, and Antimicrobial Activity against Drug-Resistant Microbes of Essential Oil from Withania frutescens L. Appl. Sci. 2021, 11, 5168. https://doi.org/10.3390/app11115168
  6. Aicha Hennia, Said Nemmiche, Susana Dandlen & Maria Graça Miguel (2019): Myrtuscommunis essential oils: insecticidal, antioxidant and antimicrobial activities: a review, Journal of Essential Oil Research, DOI: 10.1080/10412905.2019.1611672
  7. Nanasombat, S., Wimuttigosol, P. Antimicrobial and antioxidant activity of spice essential oils. Food Sci Biotechnol 20, 45–53 (2011). https://doi.org/10.1007/s10068-011-0007-8
  8. Chouhan, S.; Sharma, K.; Guleria, S. Antimicrobial Activity of Some Essential Oils—Present Status and Future Perspectives. Medicines 2017, 4, 58. https://doi.org/10.3390/medicines4030058
  9. Man, A.; Santacroce, L.; Iacob, R.; Mare, A.; Man, L. Antimicrobial Activity of Six Essential Oils Against a Group of Human Pathogens: A Comparative Study. Pathogens 2019, 8, 15. https://doi.org/10.3390/pathogens8010015

Moreover, in the manuscript published by Puvača, N.; et al. with the title “Antimicrobial Activity of Selected Essential Oils against Selected Pathogenic Bacteria: In Vitro Study.” in the Antibiotics 2021 (https://doi.org/10.3390/antibiotics10050546) there is a mention of the criteria for potency suggested by the reviewer (Sartoratto et al., Brazilian Journal of microbiology200435, 275-280), however, the authors of this manuscript have described the efficiency of the tea tree essential oils as “good antibacterial activity” although the MIC values in their experiment were recorded in the range of 2.7-6.2 mg/mL.

Point 2. Table 1, entry 13: Please double-check; this may be p-cymene. See: Romanenko & Tkachev, Chemistry of Natural Compounds200642(6), 699-701 (p-Cymene, RI = 1024; m-Cymene, RI = 1022; o-Cymene, RI = 1039). Since there are relatively large quantities of menthane monoterpenoids in TZEO, p-cymene seems more likely from a biosynthetic perspective and would agree with previously published compositions of TZEO. Table 1, entry 24: cis-p-menth-2-en-1-ol [“p” for “para”]. Table 1, entry 33: geranial is an aldehyde, not an alcohol.

Response:

The Authors would like to thank the Reviewer for the valuable comments, we agree with all suggestions, it was our overlook. Accordingly, we have introduced all suggested changes.

Point 3. Section 3.2: The discussion should be expanded. How do the DPPH and ABTS assay results compare with the values for other essential oils (i.e., how effective is TZEO as a radical-scavenging agent)? See, for example: da Silva et al., Evidence-Based Complementary and Alternative Medicine2021, Article ID 6748052. How does the antioxidant activity of TZEO compare with the major components? See: Ruberto & Baratta, Food Chemistry200069, 167-174; Sharopov et al., Natural Product Communications201510, 153-156.

Response:

The Authors are grateful for the Reviewer's constructive comments. According to the valuable suggestion we have expanded the discussion of section 3.2 in regard to the previous results for T. zygis species.

Additionally, regarding Point 3, it is a well-known fact that essential oils, as mixtures of components of different structures, mainly exert their activity due to an additive, indifferent, antagonistic, or synergistic effect of major constituents with the minor components (please see references below). In this respect, the comparison of antioxidant potency to the activity of some other essential oil (including other chemotypes of TZEO), as well as to the antioxidant activity of individual components is complicated to justify, since no straight correlation can be drawn. Herein, to express the antioxidant effectiveness of TZEO, TEAC values are provided (please see the conclusion in S. Anahi Dandlen, A. Sofia Lima, Marta D. Mendes, M. Graça Miguel, M. Leonor Faleiro, M. João Sousa, Luis G. Pedro, José G. Barroso, A. Cristina Figueiredo. Antioxidant activity of six Portuguese thyme species essential oils. https://doi.org/10.1002/ffj.1972 ).

  1. Sudipta Jena, Asit Ray, Ambika Sahoo, Pratap Chandra Panda, Sanghamitra Nayak, Deeper insight into the volatile profile of essential oil of two Curcuma species and their antioxidant and antimicrobial activities, Industrial Crops and Products, Volume 155, 2020, 112830, ISSN 0926-6690, https://doi.org/10.1016/j.indcrop.2020.112830 ;
  2. de Sousa, D.P.; Damasceno, R.O.S.; Amorati, R.; Elshabrawy, H.A.; de Castro, R.D.; Bezerra, D.P.; Nunes, V.R.V.; Gomes, R.C.; Lima, T.C. Essential Oils: Chemistry and Pharmacological Activities. Biomolecules 2023, 13, 1144. https://doi.org/10.3390/biom13071144 ;
  3. Özgür Karakaş, Fatma Matpan Bekler. Essential Oil Compositions and Antimicrobial Activities of Thymbra spicata L. var. spicata L., Lavandula X Intermedia Emeric ex Loisel., Satureja macrantha C. A. MEYER and Rosmarinus officinalis L. Article - Human and Animal Health • Braz. arch. biol. technol. 65 • 2022 https://doi.org/10.1590/1678-4324-2022210297 ;
  4. Bunse M, Daniels R, Gründemann C, Heilmann J, Kammerer DR, Keusgen M, Lindequist U, Melzig MF, Morlock GE, Schulz H, Schweiggert R, Simon M, Stintzing FC and Wink M (2022) Essential Oils as Multicomponent Mixtures and Their Potential for Human Health and Well-Being. Front. Pharmacol. 13:956541. doi: 10.3389/fphar.2022.956541;
  5. Asit Ray, Sudipta Jena, Biswabhusan Dash, Basudeba Kar, Tarun Halder, Tuhin Chatterjee, Biswajit Ghosh, Pratap Chandra Panda, Sanghamitra Nayak, Namita Mahapatra, Chemical diversity, antioxidant and antimicrobial activities of the essential oils from Indian populations of Hedychium coronarium Koen, Industrial Crops and Products, Volume 112, 2018, Pages 353-362, ISSN 0926-6690, https://doi.org/10.1016/j.indcrop.2017.12.033
  6. Leila Riahi, Myriam Elferchichi, Hanene Ghazghazi, Jed Jebali, Sana Ziadi, Chedia Aouadhi, Hnia Chograni, Yosr Zaouali, Nejia Zoghlami, Ahmed Mliki, Phytochemistry, antioxidant and antimicrobial activities of the essential oils of Mentha rotundifolia L. in Tunisia, Industrial Crops and Products, Volume 49, 2013, Pages 883-889, ISSN 0926-6690, https://doi.org/10.1016/j.indcrop.2013.06.032

Point 4. Section 3.4, Table 4: Given these data, it should be possible to determine IC50 values, which may allow for better comparison.

Response:

The Authors would like to thank the Reviewer for the constructive comments. In the described method used for determining in situ antimicrobial activity, results sometimes can vary, i.e., essential oils in the vapor phase have the possibility to exert probacterial effects in some concentrations, depending on the used oil. Considering that, the interpretation of the results in the form of IC50 values could not provide adequate information on the efficiency of tested essential oil. Below, please find some of our previous work on the matter:

  1. Valková, V.; Ďúranová, H.; Galovičová, L.; Vukovic, N.L.; Vukic, M.; Kačániová, M. In Vitro Antimicrobial Activity of Lavender, Mint, and Rosemary Essential Oils and the Effect of Their Vapours on Growth of Penicillium spp. in a Bread Model System. Molecules 2021, 26, 3859. https://doi.org/10.3390/molecules26133859
  2. Kačániová, M.; Galovičová, L.; Valková, V.; Ďuranová, H.; Štefániková, J.; Čmiková, N.; Vukic, M.; Vukovic, N.L.; Kowalczewski, P.Ł. Chemical Composition, Antioxidant, In Vitro and In Situ Antimicrobial, Antibiofilm, and Anti-Insect Activity of Cedar atlantica Essential Oil. Plants 2022, 11, 358. https://doi.org/10.3390/plants11030358
  3. Kačániová, M.; Vukovic, N.L.; Čmiková, N.; Galovičová, L.; Schwarzová, M.; Šimora, V.; Kowalczewski, P.Ł.; Kluz, M.I.; Puchalski, C.; Bakay, L.; et al. Salvia sclarea Essential Oil Chemical Composition and Biological Activities. Int. J. Mol. Sci. 2023, 24, 5179. https://doi.org/10.3390/ijms24065179

Point 5. Conclusions, lines 622-624: It is not clear how “multifold stronger” cytotoxic effect for MDA-MB-231 cells vs. MRC-5 cells can be concluded. From Figures 3 and 4, all cell lines show IC50 values > 200 ug/mL (i.e., not active according to the criteria of Cos et al.). In order to provide comparison with the normal cell line, IC50 values should be determined, and a selectivity index determined to indicate relative safety of the essential oil. The discussion in Section 3.6 should be expanded accordingly.

Response:

The Authors would like to thank the Reviewer for the constructive comments. As noted by the Reviewer, the IC50 is higher than 200µg/mL (the highest applied concentration of EO) so in discussion we have used the viability values for maximal used concentrations after long-term treatment because the strongest antiviabile effects were recorded there for all three tested cells. We have reformulated the text that was not clear regarding the differences in antiproliferative effects of EO based on the suggestions. The precise fold values of the decrease in cell viability, which represent the selectivity index, are stated in the Conclusion of the manuscript.

 Point 6. Some minor editorial corrections:

  • Lines 34-36: This sentence is repeated and should be deleted.
  • Line 88: medicine [correct the spelling]
  • Line 99: Gram-positive and Gram-negative [Gram is a proper noun and should be capitalized]

Response:

We have introduced all suggested changes. 

Reviewer 5 Report

Comments and Suggestions for Authors

Dear authors,

The manuscript entitled "Thymus zygis, valuable antimicrobial (in vitro and in situ) and antibiofilm agent with potential antiproliferative effects” describes  a  T. zygis essential oil as a beneficial and safe agent in food preservation, and its potential toxic effect on different cancer cells. It presents scientific relevance for the Food area.

However, you need to change some details/information in:

-line 111: details are needed regarding obtaining the protocol for obtaining the essential oil (yield, nature of the plant, etc.)

- figure 1 should be moved to additional materials

Comments on the Quality of English Language

Minor editing of English language required

Author Response

Reviewer #4

Dear authors,

The manuscript entitled "Thymus zygis, valuable antimicrobial (in vitro and in situ) and antibiofilm agent with potential antiproliferative effects” describes  a  T. zygis essential oil as a beneficial and safe agent in food preservation, and its potential toxic effect on different cancer cells. It presents scientific relevance for the Food area.

The Authors would like to thank the Reviewer for the time devoted for constructive and important comments to improve our paper.

However, you need to change some details/information in:

Point 1_ -line 111: details are needed regarding obtaining the protocol for obtaining the essential oil (yield, nature of the plant, etc.)

Response: As stated in the section, the essential oil tested in this study is commercially produced. Considering, those informations can be found on their website. The producer states the following information on the packaging:

Type of scent: herbal penetrating phenolic scent with a subtle citrus touch

The method of obtaining essential oil: by steam distillation of partially dried cloves

Origin of essential oil: Spain

Use and effects: significant effects against viruses, bacteria, and fungi, inflammation of the oral cavity, ear infections, muscle and joint rheumatism, sore throats, respiratory tract infections, antispasmodic effects, intestinal microflora, and digestion disorders (not long-term), nervous tension, antidepressant, fatigue.

Point 2: - figure 1 should be moved to additional materials

Response: Figure 1 shows the results of biofilm changing in different days. This figure is necessary for interpreting the obtained results, but we thank you for your suggestions.

Round 2

Reviewer 2 Report

Comments and Suggestions for Authors

The manuscript can be accepted in the present form 

Comments on the Quality of English Language

The manuscript need minor language revisions 

Author Response

Comment:

The manuscript need minor language revisions. 

Response:

According to the Reviewer's suggestion, we have revised the quality of English Language.

Reviewer 4 Report

Comments and Suggestions for Authors

I am satisfied with the authors' responses.

Round 3

Reviewer 2 Report

Comments and Suggestions for Authors

The manuscript is now accepted in the present form